# Simulated Practice Learning Experience in a Virtual Environment: An Innovative Pedagogical Approach to Practice Learning for Nursing Students

**DOI:** 10.3390/nursrep15020061

**Published:** 2025-02-08

**Authors:** Sharon Faulds, Anne Taylor

**Affiliations:** Faculty of Health Sciences and Sport, University of Stirling, Stirling FK9 4LA, UK; a.d.taylor@stir.ac.uk

**Keywords:** innovation, pedagogy, simulated practice learning experience, virtual practice learning, student nurse, practice placement, virtual learning, nursing education, online learning

## Abstract

**Background/Objectives**: The use of simulated learning as a teaching approach has been used and embedded in nursing theoretical curriculum for many years. There is a wealth of evidence to support the positive impact simulated learning or simulated-based education can have on the student experience, developing skill competency and enhancing patient outcomes. However, the evidence on the use of simulation as a replacement for clinical practice learning in undergraduate nursing education is limited. In response to the challenges posed by the COVID-19 pandemic, the authors introduced virtual simulated practice learning experiences (SPLE) for a cohort of year one pre-registration adult and mental health nursing students. The SPLE project aimed to assess the effectiveness of simulated practice learning as a viable alternative to traditional clinical practice learning and to explore student satisfaction with the new practice learning experience approach. **Methods:** All year one student nurses attending the four simulated practice learning experience (SPLE) weeks were invited to participate and complete a generated questionnaire within the virtual practice environment on their final day of each SPLE week. The questionnaire employed a mix of both quantitative and qualitative questions across key areas to evaluate the effectiveness of the SPLE and explore student satisfaction with their overall practice learning experience. **Results**: A total of 216 students participated in the simulated practice learning weeks across the spring semester in 2023 with a response rate of 98–100% across all four SPLE weeks. Students reported an overall satisfaction score of 88%, highlighting their preparedness and positive feedback on the organisation, delivery, and content of the SPLE. Qualitative analysis revealed key themes, including the development of transferable skills and personal growth, the value of peer learning, the benefits of a virtual environment, and appreciation of service user and healthcare professional input. Students reported significant personal growth, improved communication skills, and a deeper understanding of holistic care through interactive and collaborative learning experiences. **Conclusions:** This evaluation underscores the innovative potential of simulated practice learning to enhance nursing practice education, emphasising the importance of integrating emerging technologies and diverse pedagogical approaches. The findings suggest that SPLEs can effectively prepare nursing students for the complexities of clinical practice while addressing the evolving demands of healthcare. Future research should focus on longitudinal studies to assess the sustained impact of simulated learning on clinical experiences and professional development.

## 1. Introduction

Innovation is not just about new technologies or methodologies but also about adopting a mindset that encourages creativity, problem-solving, and the implementation of new ideas in nurse education and in clinical practice [1]. The COVID-19 pandemic and ongoing NHS service re-design has forced many nurse educators to be innovative and creative to maintain practice learning placements for student nurses, with many United Kingdom Higher Education Institutions (HEI) adopting virtual or simulated practice learning experiences [2]. This novel and unique approach to practice learning provided pre-registration nursing students with a means to fulfilling the required Nursing and Midwifery Council (NMC) practice hours allowing them to successfully complete their nursing programme on time, attain registration, and furthermore join the much-needed nursing workforce.

### Background

In the United Kingdom, the Nursing and Midwifery Council [3] plays a crucial role in setting and maintaining the standards for undergraduate nursing education and practice. To ensure high-quality care, the NMC expects nursing students to effectively integrate theory into their direct practice learning, accruing 2300 hours during their nursing programme for registration. Due to the significant effect of COVID-19 on practice learning, in 2021, the NMC approved the continued use of the COVID-19 recovery standards permitting HEIs to deliver 300 to 600 hours of practice learning using a range of new and innovative practice simulation methods. Simulated practice learning, defined by the NMC, is practice learning that meets requirements set out in the NMC standards around practice learning, in particular the requirements contained within the standards for pre-registration nursing programmes and the standards for student supervision and assessment [4]. Any approaches or methods adopted for practice learning through the development of simulated practice learning experiences (SPLEs) must align with the desired practice learning outcomes and contribute to the student being able to demonstrate safe and effective practice [2].

The use of simulated learning as a teaching approach has been used and embedded in nursing theoretical curriculum for many years. There is a wealth of evidence to support the positive impact simulated learning or simulated-based education can have on the student experience, developing skill competency and enhancing patient outcomes [5,6,7]. The use of simulated or virtual learning aims to provide nursing students with an authentic and immersive learning environment, enabling the application of theoretical knowledge to practical scenarios [8]. This experiential learning approach allows students to develop their critical thinking skills and bridge the gap between theory and practice; however, it is critical that the learning environment is authentic and is representative of real-life situations to promote fidelity and authentic learning for the students [6]. The change to the NMC standards in 2021 allowed HEIs in the UK to consider how they could replicate the learning from the theoretical training of nursing students into the practice learning context and accrue direct practice hours.

Research into the use of simulation as a replacement for clinical practice learning in undergraduate nursing education is limited [9]. By scoping the literature, it is evident the value that virtual or simulated practice learning can have on students from across a range of health professions. Pit et al. [10] emphasises, from a medical student’s perspective, how virtual practice learning allows the students interest in healthcare technology to grow, which stems from its increasing conventional relevance and its invaluable role in their future careers. This view is reinforced by studies from West [11] and Mian and Khan [12] who emphasise the importance of integrating digital technologies into medical education. They argue that this is not only a necessary response in light of the COVID-19 pandemic but also a critical foundation for future educational practises in healthcare. Inman et al. [13] highlight the effectiveness of technology in facilitating reflective sessions and supervision, promoting flexibility, efficient time management, and, moreover, access to a wider team of supervisors. The use of virtual or digital technologies is regarded as essential for enhancing learning and upholding professional standards and students should expect to be involved in simulation to prepare them for working in clinical environments increasingly using technology. Medical students appreciated the exposure to a broad range of virtual clinics, diverse clinicians, and peers, thereby significantly enhancing their practice learning capacity [14], with Peart et al. [15] demonstrating that virtual practice learning offers unique opportunities for students, fostering flexibility and lateral thinking. However, there are also legitimate concerns and challenges associated with this mode of learning reported in the literature. Triemstra et al. [16] reports that in-person practice learning offers many more benefits than virtual practice learning in terms of collaborative practice, professional identity formation, and the accumulation of clinical knowledge and judgement. This consensus is shared by Hammond et al. [17], emphasising that patient contact is essential for developing context-specific communication skills and the irreplaceable nature of face-to-face learning experiences. Furthermore, Walker and Stapleton [18] report that no medical student preferred virtual learning, citing reduced opportunities for meaningful patient contact and less engaging experiences, with Franklin et al. [19] finding that medical students felt inadequately prepared for practice, leading to concerns about future career prospects. A recent scoping review carried out by Amankwaa et al. [20] presents a wealth of evidence on the impact on the use of virtual and simulated approaches adopted by nurse educators in HEIs during the COVID-19 pandemic. They highlight the negative impact on nursing student mental health and well-being [21], the effects of blended learning [22], and the challenges of online learning [23]. The authors describe how some HEIs adopted simulated or virtual approaches to clinical placements using simulated skills sessions, experiential case studies and tele-health technology, noting the many benefits of these approaches in the delivery of innovations in a range of clinical environments. However, they note the lack of studies that demonstrate the long-term effectiveness and sustainability of virtual practice learning as we move away from the pandemic. This paper will describe and present the evaluation from the first run of a novel simulated practice learning experience project, showing its effectiveness on practice learning and satisfaction with year one pre-registration nursing students.

## 2. Materials and Methods

### 2.1. Design and Context of the SPLE Weeks

To maximise the opportunity that the NMC had provided, the project team reviewed clinical learning environments that students typically had limited or no access to during their nurse training or where placement capacity post-pandemic had significantly reduced. In response, the team designed the four-week SPLE to address these gaps, ensuring students gained exposure to crucial areas of practice—see Figure 1. Historically, adult and mental health student nurses in maternity care placements would have two weeks observing and shadowing a range of midwives across various clinical settings, learning about their role. To reflect this observational experience, a maternity SPLE week was developed, allowing the students to follow a fictional couple’s pregnancy journey from ante-natal to post-natal care, gaining a unique insight into the clinical needs and care of a pregnant woman and her family. Currently only, a small number of mental health students gain access to direct practice placements in the field of learning disability nursing. The project team had a unique opportunity to develop an SPLE to immerse all students, providing them with the opportunity to engage with key healthcare professionals and individuals with learning disabilities, gaining insights into real-life challenges and care needs in a way that was previously inaccessible. A child health SPLE week was developed which provided a unique and innovative opportunity for students to gain exposure to caring for individuals from birth through to adolescence, an experience that had previously been limited due to the shortage of child health practice placements. Finally, a community care SPLE week was developed that offered a solution to the post-pandemic challenge of limited placement opportunities within district nursing community teams for the students. All students during this SPLE week learned how district nursing teams assess, discuss, and plan compassionate, dignified care for patients, an essential skill for future nurses and engaged in activities such as completing referral forms, practising assessments, and learning procedural skills like using syringe drivers.

Each SPLE week was aligned with the desired practice learning outcomes for year one students and mapped to key platforms and proficiencies, ensuring the student was able to demonstrate safe and effective practice through engaging in key activities and assessment. Scenarios and case studies for each of the SPLE weeks were drawn from real clinical experiences and carefully aligned with the NMC Standards of Proficiency for Registered Nurses [24], ensuring an authentic and immersive learning environment. These scenarios allowed students to follow a cohesive and engaging narrative that spanned various aspects of holistic care across the four SPLE weeks. In the maternity week, students followed the fictional couple’s pregnancy journey, from giving birth to a healthy baby to experiencing the birth and care of a child with a learning disability during the learning disability week. The child health week expanded on this narrative by exploring the role of the health visitor in supporting a family with both a healthy child and a child with mental health or complex needs. Finally, the community care week introduced a neighbour of the fictional couple who required ongoing support from district nursing teams, further emphasising the continuum of care, from birth to death, across the four SPLE weeks. These various learning experiences allowed students to consolidate their clinical learning in a safe and supportive virtual environment.

Canvas by Instructure, the universities web-based learning management system, was utilised to host the SPLE required educational materials, blending technology with pedagogy to create an immersive and interactive learning environment [25]. Beyond its conventional use, Canvas was strategically operated to host all SPLE-required educational materials, ensuring seamless access to weekly programmes, daily itineraries, Microsoft Teams links for live clinical sessions, weblinks, quizzes, and multimedia content such as videos and podcasts. This approach transformed Canvas into a dynamic, central hub for students, enabling a cohesive and integrated learning experience. The Information Technology team played a crucial role in customising Canvas to support the unique needs of the SPLE weeks and supporting the facilitation of each week, significantly contributing to the smooth operation of the IT platform. The platform’s user-friendly interface and comprehensive toolset made it an ideal choice for delivering the SPLE’s content, demonstrating a forward-thinking approach to practice learning delivery in a virtual environment [26].

Thinglink (Version K12) was embedded and utilised in three out of the four SPLE weeks to provide an authentic clinical learning environment. Thinglink is a digital platform that creates visual interactive tools which allow users to turn any image or video into an interactive and visual learning experience [27]. Its interface is user friendly, intuitive, and incorporates immersive reader, which supports accessible and inclusive learning. The Thinglink platform permits the user to upload a base image and then add a selection of icon tags, which, once clicked, can provide links to additional text, images, audio/media files, or links to websites or documents. These icon tags can be selected in various symbol formats or colours and placed anywhere within the base image [28]. The final product is a flexible and navigable resource with multiple media sources linked together in a logical manner.

As the SPLE was required to meet the standards for student supervision and assessment (SSSA) [29], a practice “virtual induction” was conducted on the first morning of each SPLE week to familiarise the students with the Canvas virtual practice environment. The students completed bespoke SPLE pre-practice activities to ensure they all had access to the digital and online resources they would need and furthermore were informed via the virtual induction how to access support available to them in case of any issues. This session guided students through activities to ensure their devices were suitably prepared to accommodate the technical requirements of the SPLE, such as testing software compatibility and internet connectivity. This comprehensive preparation aimed to equip students with the necessary knowledge and confidence to navigate the SPLE successfully.

Daily debriefs played a pivotal role in enriching the simulated practice learning experience by promoting reflection, collaboration, and knowledge consolidation among the students during and at the end of each day. Utilising Microsoft Teams, these debrief sessions brought together a diverse learning community in the virtual environment, enabling students to share their experiences, receive real-time feedback from lecturers and clinicians, and engage with service users in a supportive and interactive setting.

A structured approach to engagement and assessment was implemented through the Canvas platform to support the students in demonstrating their learning. This was crucial in meeting the standards for student supervision and assessment [29] and ensuring that all practice hours were monitored and accounted for in line with NMC requirements. To ensure attendance and participation were accurately tracked, students were required to demonstrate their attendance at the beginning, middle, and end of each day through Microsoft Teams calls, with registers taken to cross-check their hours. This structured attendance system not only supported accountability but also helped maintain engagement throughout the day. Additionally, at the end of each day, students completed an online quiz or activity specifically designed to assess their understanding of that day’s learning materials, further ensuring that their knowledge and skills aligned to the expected learning outcomes. Upon completing the mandatory end-of-day activity, the students then generated and downloaded a certificate as evidence of their engagement and knowledge. These certificates (five per week) were then uploaded to a dedicated section within their electronic Practice Assessment Document (ePAD). Once reviewed by the students’ nominated practice supervisor, who was a clinician involved in the specific SPLE week or a member of staff from the SPLE team, the allocated practice hours for that day (maximum 6 hours per SPLE day) were awarded. This method of assessment not only kept students actively engaged but also provided a clear and documented way to track their learning progress and practice hours, ensuring a systematic process was in place to meet NMC governance.

### 2.2. Study Design and Methodology

The objective of this project evaluation was to establish the effectiveness of the SPLE and to explore student satisfaction with their simulated practice learning experience.

An evaluation questionnaire was used to gather a more comprehensive understanding of the effectiveness of the SPLE and the students’ experiences and perceptions of the SPLE weeks. The questionnaire, which integrated both qualitative and quantitative questions, offered a thorough understanding of the SPLE project’s impact, implementation, and context. It allowed the ability for the project team to compare the findings from the questionnaire, thereby enhancing the validity and reliability of the project evaluation. Additionally, using a mix of questions during the evaluation allowed for explanations that might be overlooked by quantitative data alone, providing a more holistic perspective on the novel SPLE project [30].

Fitzpatrick et al. [31] describe project evaluation as a systematic methodology used to assess the effectiveness, efficiency, and impact of a given project. It involves the collection and analysis of both qualitative and quantitative data to measure outcomes against predefined objectives and goals. To collect the data in this project, an evaluation questionnaire was designed and embedded into Canvas. The questionnaire consisted of 16 questions that aimed to assess key aspects of the students’ overall experience, learning, and activities, and their perception of the virtual practice learning environment. The student evaluation questionnaire was categorised into key questions relating to the specific SPLE content, information, and use of technology, and a shortened version of the validated NHS Education for Scotland [32] Student Practice Learning Experience Feedback form was included, which is the standard practice placement evaluation that is completed by all students across all practice placements. By completing the evaluation questionnaire, students enabled the project team to capture valuable insights into their learning and engagement during the SPLE and identify ways to enhance and improve the SPLE weeks in future iterations. Eleven of the questions were quantitative in design to provide numerical data that could be analysed statistically to identify trends and patterns. The responses on the questionnaire were generated using the quiz function embedded within Canvas platform using the Likert scale of strongly agree; agree; neither agree or disagree; disagree; and strongly disagree. This scale allowed for the quantification of subjective opinions, enabling the analysis of respondents’ levels of agreement or disagreement with specific statements [33]. The remaining five questions were open-ended to obtain qualitative feedback on student experiences. These questions allowed students to provide detailed and descriptive responses, offering insight into their personal experiences of the SPLE weeks, any challenges they experienced, and any suggestions on how the SPLE weeks could be improved in future iterations. The qualitative data gathered through these open-ended questions provided the project team with an understanding of the students’ overall perspectives and allowed for a more nuanced analysis.

Utilising Canvas analytics, the team were able to efficiently export the student evaluation responses and download it into a Microsoft Excel file. By pulling descriptive statistics from the excel file, the team were able to showcase the analysis of respondents’ levels of agreement or disagreement with specific statements, offering valuable insights into the students’ perceptions and potential areas for improvement in future iterations. Drawing on Braun and Clark [34,35] as a foundation, thematic analysis of the data from the open-ended questions was conducted, adhering to the distinct stages of analysis. Thematic analysis was chosen due to its theoretical freedom, providing a flexible and useful research tool, enabling one of the researchers (SF) to present a rich and detailed account of the data. Initially the student responses were read multiple times to foster familiarity with the data while simultaneously noting down early themes and ideas. Subsequently, the dataset was coded to identify noteworthy characteristics within the data. Finally, a comprehensive interpretation of the data were synthesised through the generation of themes. The ability to easily access, export, and analyse these data ensured that the analytical process was both thorough and efficient, leading to a summary of the key patterns and themes from the student responses.

### 2.3. Ethical Implications

The SPLE project was not designed as a primary research study; hence, no formal ethics was sought or approved from the institutions ethical committee. At the time the project was developed, the primary focus was on reducing the impact of COVID-19 on year one students’ clinical practice hours, and data were collected via the questionnaire evaluation to establish if the SPLE was effective and to explore student satisfaction. Although ethical approval was not sought and written informed consent was not obtained from all students, they were informed verbally by the project lead (AT) on numerous occasions that an analysis of the evaluation data would be generated to produce a project report that would be discussed and presented to faculty and disseminated to our clinical partners.

The final evaluation questionnaire was set up as a quiz and the data were downloaded in an excel format. This ensured that all students responses were fully anonymised, and the excel sheet contained no identifying data safeguarding the student’s privacy and individual responses could not be traced, ensuring the protection of students’ rights, dignity, and overall well-being. Additionally, the students were free to choose how many questions they answered and the level of detail they wished to provide as not all questions needed to be answered. The students were not coerced into completing the full evaluation and had freedom to choose which questions they wished to answer. An announcement was sent to the cohort of students informing them of future publications and if any student had any queries/questions on the use of their data to contact the project lead. No students contacted the project lead.

## 3. Results

### 3.1. Quantitative

A total of 216 students participated in the simulated practice learning weeks across the spring semester in 2023 with a response rate of 98–100% across all four SPLE weeks. Student numbers attending each week ranged from 198 to 216 (see Table 1) with the variation in the sample size per week accounting for sickness, non-engagement, and students taking a break from their programme of study.

The overall satisfaction from the four-week SPLE evaluation yielded positive results. When combining the responses of “agree” (35%) and “strongly agree” (53%), the total satisfaction score by the students reached 88% (see Figure 2) with Figure 3 illustrating the satisfaction in response rates for each individual SPLE week. Students appeared to be most satisfied with the maternity week and less satisfied with the child health week.

On average, across all SPLE weeks, 92% of students either agreed or strongly agreed that the virtual space was easy to access and navigate, reflecting the platform’s user-friendliness and effectiveness in facilitating their practice learning experience—see Figure 4.

Additionally, 95% of students either agreed or strongly agreed that participating in the SPLE weeks significantly enhanced their knowledge of that specialist field of nursing, with learning disability being a popular choice with the students—see Figure 5.

The high response rate indicates that students fully engaged in the simulated practice learning experience weeks. Student responses suggest that overall, they were satisfied with the SPLE as a replacement for practice learning. The positive results suggest the content and organisation of the week highlights the success of both the virtual platform and the SPLE content in advancing students’ practice learning development and practical understanding. Furthermore, the results suggest that the content of the SPLE was appropriately aligned with the practice learning outcomes and the material was relevant and accessible for the students. The maternity week appeared to be more popular with the students, which could be explained with the first use of Thinglink in this week. The immersive environment provided by this platform offering a fresh and interactive perspective on the various stages of maternity care could be a factor in this result. The positive results on the overall organisation and delivery of the SPLE suggest that each week was structured and executed in a manner that appears to have met students’ practice learning needs and expectations.

### 3.2. Qualitative

The qualitative responses from the questionnaire showed several key themes that reflected student experiences and perceptions of the SPLE. The themes were categorised into four main headings: transferable skills and personal growth; the value of peer learning; the benefits of learning within the virtual environment; appreciation of service user; and healthcare professional’s input. The findings demonstrate the fundamental value and impact students experienced in fostering essential skills and facilitating collaborative learning experiences from the SPLE.

#### 3.2.1. Transferable Skills and Personal Growth 

Many students reported how the SPLE was pivotal to their personal growth and preparation for upcoming direct practice learning placements, attributing their sense of readiness and confidence to the comprehensive nature of the SPLE:

“*I enjoyed this week very much. My knowledge of maternity care has been developed to a satisfactory level. Thus, I feel more confident to apply these skills as a future nurse caring for new mothers and their partners and supporting them with their questions*”.

“*I feel it was very beneficial to learn about maternity care, as made me aware of so much more than just the role of the midwife in maternity care, but how pregnancy can impact on the other branches of nursing too*”.

“*I felt that I gained so much knowledge which I will be able to take forward into practice and help me in my personal life with my family*”.

“*I enjoyed how interactive it was, allowing us to take the information and apply it to the referral forms and care plans*”.

#### 3.2.2. Peer Learning

Many of the student responses showcased the benefits of peer learning through small group discussion and activities plus the debrief sessions during the day with clinical staff and service users:

“*The drop in’s and end of day discussions each day were extremely helpful hearing other students’ opinions and questions on the material we were learning that day*”.

“*I enjoyed having the opportunity to interact with the other students and lecturers. The days were easy to follow and had very interesting chats*”.

“*The end of the day discussion was valuable as it rounded off the learning and gave opportunities to ask questions*”.

“*I enjoyed being able to see things from a different perspective. This allowed me to challenge my knowledge and help me think on the spot/problem solve*”.

#### 3.2.3. Benefits of Learning Within the Virtual Environment

Considering virtual placements are a relatively novel concept in the field of nursing education, it was reassuring to receive student feedback reporting their perceived benefits of virtual learning. The students’ reported their experience was notably enhanced by the diverse range of learning methods employed, including videos, live sessions, and interactive tasks. They highly praised the interactive nature of the sessions and the inclusive learning environment, which encouraged engagement through group work, lively discussions, and a supportive learning community. The students also valued the opportunity to collaborate and share perspectives and found engaging with both lecturers and healthcare professionals from various disciplines particularly enriching. This multifaceted interaction not only provided real-world insights but also made their learning process dynamic and engaging:

“*I really enjoyed all parts of it. At first, I was unsure how well virtual placement would be with it being online but by the end of the first day I thought it was amazing and worthwhile. The lecturers definitely made you want to go research further as you can tell they are so passionate about it*”.

“*I enjoyed working in groups as it allowed us to communicate in different ways and build confidence*”.

“*I truly enjoyed SPLE weeks. The content was good, with excellent explanations, and the materials were accessible and informative*”.

“*I liked the layout of having a lot of live calls with the lecturers and staff, it made it feel very inclusive and you felt a part of everything*”.

#### 3.2.4. Appreciation of Service User and Healthcare Professional Input

The students highly valued the opportunity to hear personal stories and real-life experiences from guest speakers, families, and professionals, particularly in the learning disability sector. These narratives provided deep insights and encouraged emotional connections, significantly enhancing their understanding and empathy. The real-world perspectives shared by these individuals offered practical knowledge and contextual understanding, which was deemed invaluable in preparing the students for real-life applications in their future training. By engaging in these activities, students developed a deeper understanding of how theoretical knowledge underpins practical skills and decisions, thereby enhancing their overall competence and confidence in their field

“*The participation of the families, speech therapist, learning disability nurses, and their information helped to improve my skills*”.

“*I particularly enjoyed listening to the CALMS meeting as we got to see a real-life scenario*”.

“*The guest speaker was great, her shared experience was more than a book I could ever have read*”.

“*I really appreciated the real-life experiences we had from parents, learning disability nurses, and speech and language therapists*”.

#### 3.2.5. Areas for Improvement

While students recognised the value of the SPLE’s organisation, delivery, and content, they also provided feedback highlighting areas for improvement. It was acknowledged that differing student learning approaches made it challenging to allocate a set timeframe for tasks that suited everyone. A small number of students commented on the pacing, with remarks such as:

“*If anything, a little more content would have been good, however I do understand it was like this purposefully not to overwhelm students*”.

“*Lots of information within a short period*”.

Some students felt that the actual practice experience would have been more beneficial. As one student noted:

“*As much as I enjoyed the information, I personally think being on a maternity ward or being on an actual placement would have been more beneficial*”.

Additionally, a few students found navigating the virtual space stressful, as one student shared:

“*It was only the first day and was so overwhelming with information overload and instructions coming from lectures over team meetings. The attempt to navigate my way through the computer and learn was stressful and tense for that day…*”.

With another student suggesting more flexibility in the timing of activities:

“*Have opportunity to complete quizzes earlier, but understand it for timing of days*”.

This feedback reflects the varying educational support needs of students and emphasises the importance of considering diverse learning preferences when designing future iterations of the SPLE.

## 4. Discussion

As discussed in the background, there is limited research specifically focusing on simulated practice learning in a virtual environment in nursing, with insights from other healthcare practitioners providing valuable perspectives on the benefits and challenges associated with virtual clinical placements. The COVID-19 pandemic and its impact on the healthcare system led to disruptions in practice learning, prompting the NMC [2] to modify their standards to incorporate virtual approaches. This pushed nurse educators to be innovative and creative to maintain practice learning placements for student nurses, with many United Kingdom HEIs adopting virtual or simulated practice learning experiences [2]. This was a point in time where nurse educators needed to adopt a mindset that encouraged creativity, problem-solving, and the implementation of new ideas, as highlighted by Leary et al. [1]. The aim of this evaluation was to explore if simulated practice learning experience could offer an effective viable alternative to conventional student nurse practice learning placements and what impact it would have on student satisfaction.

The quantitative findings suggests that students overall had a positive experience and were satisfied with their simulated practice learning placements, with 92% agreeing that the virtual platform was easy to access and navigate, highlighting its user-friendliness. Additionally, 95% reported significant knowledge enhancement in their specialist nursing fields, showing that SPLE could be a viable alternative to conventional practice placements. Students reported their satisfaction with the content and organisation, which aligned with practice learning outcomes and provided relevant and accessible practice material. The maternity week was popular possibly due to the immersive nature of Thinglink, offering an interactive perspective on maternity care. Overall, the results indicate that the SPLE weeks were effectively structured to meet students’ learning needs, reflecting the success of the platform and content in advancing their practice learning development. It is encouraging to note that these results reflect recent findings by Holt [4] who found that students were satisfied with their simulated practice learning experiences if the SPLE material and environment was organised, included people who use services and their carers, and had students work with authentic and “real life” clinical scenarios. Student satisfaction was rated high due to the equitable nature of the SPLE, allowing all students the opportunity to access practice learning from a diverse range of experiences not available in conventional practice placements which is evident SPLE project.

The qualitative findings are also consistent with similar results documented in the literature, where students identified the following themes that helped them feel prepared for future practice placements.

### 4.1. Transferable Skills and Personal Growth

Billett [36] highlights the importance of transferable skills in the integration of learning experiences across educational and practical settings, emphasising their necessity for student nurses to effectively transition into professional practice. Despite being in year one of the programme, the students were able to acknowledge the significance of transferable skills in nursing. They recognised the importance of these skills due to the diverse nature of nursing and the array of challenges that nurses face in practice. The SPLE played a pivotal role in their personal growth, significantly improving their communication abilities and deepening their grasp of holistic care approaches, thereby enhancing their readiness to tackle the complex demands of nursing practice. The virtual placement was particularly valued for its practical application of theoretical knowledge. Students found activities such as completing referral forms and conducting assessments highly beneficial. These hands-on experiences not only reinforced their learning but also enabled students to consolidate their knowledge and develop crucial skills for their future roles in clinical practice. Practice placements offer students “hands on” practical skill development, exposure to real healthcare settings, interprofessional collaboration, and, furthermore, patient interaction. The SPLE’s were instrumental in preparing them for their future nursing placements, equipping them with the necessary knowledge, skills, and attributes to deliver safe and effective patient care [37] while simultaneously facilitating their personal growth as emerging professionals.

### 4.2. Peer Learning

The value of peer learning through group work was found to be a benefit of the SPLE. The students discussed how this allowed them to consider aspects of care and hear varying perspectives which encouraged them to consider their own values, create a supportive learning environment that inspires collaboration, enhances skill acquisition, and, furthermore, prepare for the collaborative nature of healthcare work [38]. It adopts a supportive learning community, encourages diverse perspectives, and prepares individuals for teamwork in nursing practice [39]. By leveraging the power of peers, nursing students and professionals can enhance their learning experiences, develop essential skills, and contribute to their professional growth and development. Amankwaa et al. [20] emphasise that learner engagement approaches, such as blended learning and small group discussions, are vital for establishing knowledge acquisition in the online learning environment. These methods facilitate interaction and collaboration among students, which are essential for effective learning. However, they also note that many papers in the review merely described the transition of face-to-face teaching content to an online format without detailing the necessary modifications to ensure student engagement. This suggests that simply replicating in-person teaching online may not be sufficient for effective learning. This was closely linked to the importance students placed on the daily debrief sessions during the SPLE weeks, which provided a safe space for them to consolidate their learning in the safe presence of their peers, maximising educational benefits [20].

### 4.3. Benefits of Learning Within the Virtual Environment

Students expressed a growing interest in healthcare technology and recognised its relevance to their future careers [10]. Triemstra et al. [16] and Twogood et al. [14] report that virtual placements offer student flexibility, eliminate geographical restrictions, provide exposure to diverse clinicians, and enhance critical thinking skills. These insights from the literature align with the findings from this project evaluation where students positively expressed the flexibility and control they had over their own learning process, particularly with the ability to work at their own pace and, furthermore, expressed how they enjoyed the possibility of independent learning within a comfortable environment. It is, therefore, evident that the students appreciate the immersive SPLE learning environment and the ability to apply theoretical knowledge to virtual scenarios.

### 4.4. Appreciation of Service User and Healthcare Professional Input

Service user involvement in nursing education not only enriches the learning experience but also prepares the students to become more effective and compassionate healthcare providers [40]. The students’ appreciation for the input of service users and healthcare professionals stands out as a critical component of their simulated practice learning experience. Engaging with personal stories and real-life scenarios, particularly in the learning disability SPLE, offered students a profound emotional connection and a deeper understanding of the complexities involved in patient care [41]. The firsthand accounts from families, speech therapists, and learning disability nurses were highly valued as they provided practical, real-world insights that textbooks alone could not offer. These interactions helped bridge the gap between theory and practice, enriching students’ empathy, which is essential in healthcare. As students reflected on these experiences, they highlighted how this exposure enhanced their understanding of how theoretical knowledge is applied in clinical decision-making and skill development. This opportunity not only boosted their confidence but will prepare them for future professional challenges. Student feedback emphasised that the participation of guest speakers and professionals added significant value to the simulated practice learning environment, making the content both relevant and impactful.

### 4.5. Areas for Improvement

While students recognised the value of the SPLE’s organisation, delivery, and content, their feedback also highlighted areas for improvement, reflecting some challenges discussed in the literature. Virtual practice learning has been associated with anxiety, motivation concerns, social isolation, and mental health issues [42,43]. Similarly, a small number of students in this project evaluation reported difficulties, such as navigating the virtual space and adjusting to the pacing and structure of activities. For instance, some students found the first day overwhelming, citing “information overload” and “stressful” attempts to adapt to the virtual format. This aligns with Poon et al. [44] who noted that adapting to new learning formats can increase stress levels.

Additionally, some students expressed a preference for in-person practice learning, echoing findings from other studies that emphasise the benefits of meaningful patient contact [16,18]. As one student commented, “*Being on an actual placement would have been more beneficial*”. This reflects ongoing concerns about the limitations of virtual placements in promoting hands-on clinical experience.

However, while these concerns are valid, the SPLE evaluation suggests that many initial uncertainties about the online format were mitigated by its clear organisation, diverse resources, and innovative tools like Thinglink. Students praised the use of interactive elements, particularly during the maternity care week, which provided a realistic authentic clinical experience.

These findings highlight the importance of student feedback in future iterations to address both common and context-specific challenges of SPLE, ensuring that virtual placements are inclusive, adaptable, and responsive to varying student needs.

### 4.6. Strengths

The project evaluation has effectively identified numerous advantages of SPLE. One of the key strengths lies in the collection of data from a large cohort of students, which adds significant credibility to the findings and reinforces the reliability of the conclusions drawn. The use of novel and unique methods and tools to enhance practice learning has been particularly noteworthy, demonstrating a commitment to innovative educational approaches.

Furthermore, the SPLE ensured equity across all students as each participant was exposed to the same level of learning and practice opportunities, promoting an inclusive practice education environment supported by Holt [4]. As recommended in the scoping review by Amankawaa et al. [20], the incorporation of debriefing sessions at the end of each day provided a safe space for students to reflect on their experiences, guided by experienced clinicians, which enhanced their learning and professional development. Additionally, the emphasis on group work and communication encouraged collaboration among students, helping them to develop essential teamwork skills that are critical in healthcare settings.

The sustainability of the SPLE is further demonstrated by its successful validation by the NMC in 2024, leading to its integration into the Adult and Mental Health BSc Nursing programmes. This achievement reinforces its effectiveness as a long-term practice learning platform, providing a comprehensive and supportive learning experience for students. Together, these strengths underscore the SPLE’s value as a recognised component of nursing education, ensuring its ongoing contribution to student practice learning. Each iteration of the SPLE will require the team to review and refine each category of feedback questions to maintain effectiveness and ensure student satisfaction.

### 4.7. Limitations

The findings are based solely from the feedback provided by first-year students. Therefore, the conclusions drawn cannot be generalised to represent the opinions of student nurses from all years across nursing programmes. The evaluation questionnaire was designed and developed for this project and was only piloted with two year four students.

### 4.8. Future Considerations and Recommendations

The positive outcomes from this evaluation reinforce the importance of influencing emerging technologies to sustain the effective design and structure of future simulated practice learning experiences. Through simulation, augmented reality immersive technologies, and the virtual shadowing of clinicians and service users, students can engage in more dynamic and interactive learning experiences. These innovations not only enhance critical thinking and bridge the theory–practice gap but also create flexible, scalable solutions for training nurses to meet the evolving demands of healthcare. By harnessing these tools collectively, we can further ensure students are well equipped to provide safe, compassionate care in real-world settings [45].

Nevertheless, the long-term impact of SPLEs remains underexplored due to the varying placement areas assigned to students. Future research should focus on how simulated practice learning influences students’ clinical experiences over time. By examining future cohorts through longitudinal studies, we can assess the sustained effectiveness of SPLEs and adapt these innovations to continuously improve nursing education. This iterative approach not only strengthens practice education but also ensures it remains at the forefront of technological advancements in healthcare training.

## 5. Conclusions

In summary, this project evaluation has proven that simulated practice learning experiences can be effective and students are satisfied with their practice learning experience. The findings underscore both the benefits and challenges of simulated practice learning placements with the positive outcomes of the acquisition of transferable skills, the value of peer learning, and the appreciation of input from service users and healthcare professionals demonstrating the effectiveness of virtual environments for enhancing practice learning.

As the landscape of nursing evolves, simulated placements offer unique opportunities for flexibility and access to diverse practice experiences. This forward-thinking approach not only prepares students for the immediate demands of their roles but also cultivates critical thinking and adaptability for their future careers in healthcare. Moving forward, it is essential to maintain this momentum of innovation, ensuring that the practice education strategies employed not only address current needs but also anticipate future developments in the field of nursing education. By integrating both technological advancements and key elements of traditional practice, we can better prepare future healthcare professionals to meet the growing demands of the profession and deliver safe, compassionate care in a rapidly changing healthcare setting.

The second iteration of the SPLE (2024) will further refine and build on these strengths, focusing on sustainability and continuous improvement.

## Figures and Tables

**Figure 1 nursrep-15-00061-f001:**
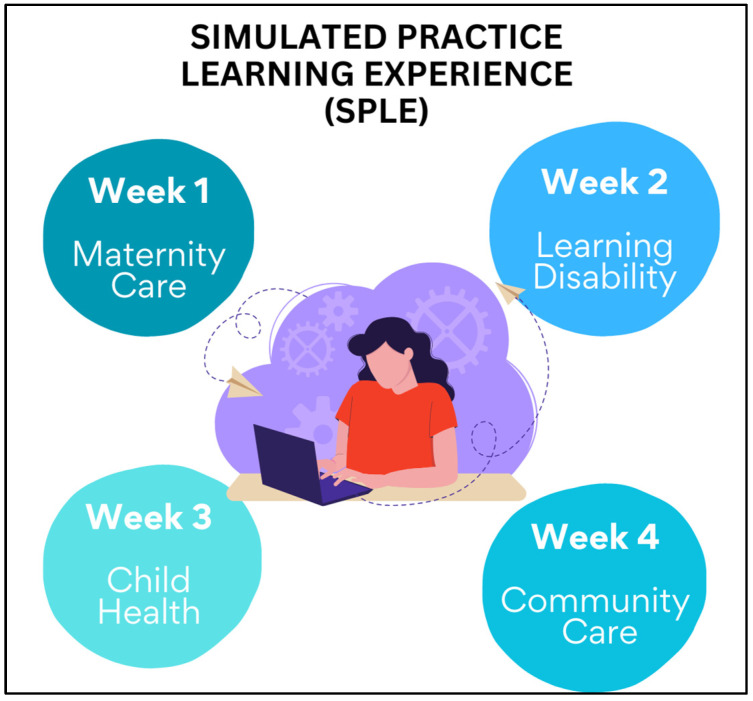
Simulated practice learning experience (SPLE) 4-week design.

**Figure 2 nursrep-15-00061-f002:**
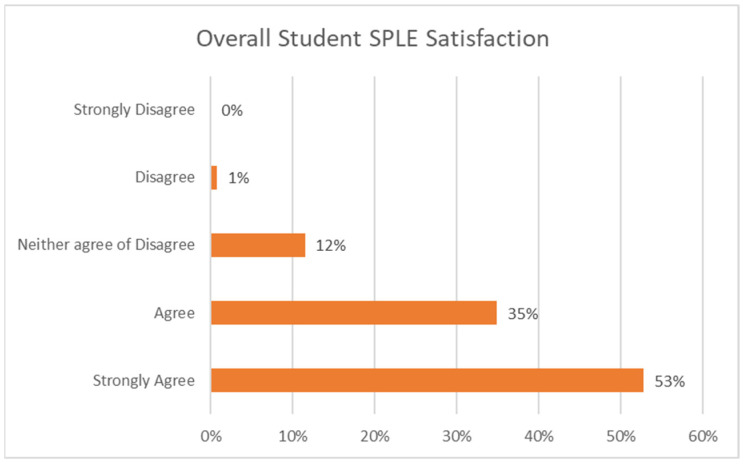
Overall student SPLE satisfaction graph.

**Figure 3 nursrep-15-00061-f003:**
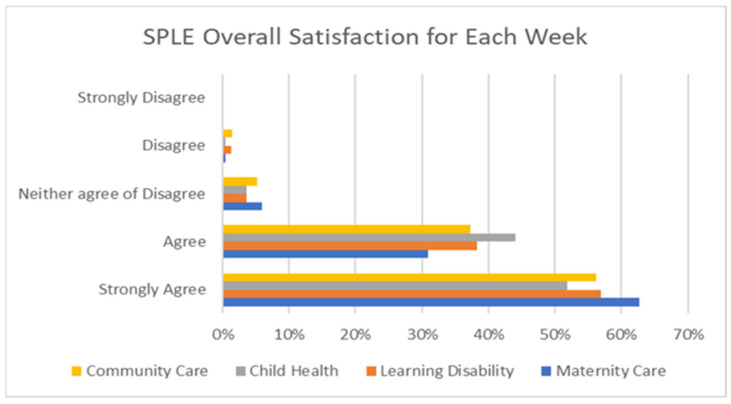
Overall student satisfaction for each individual SPLE week graph.

**Figure 4 nursrep-15-00061-f004:**
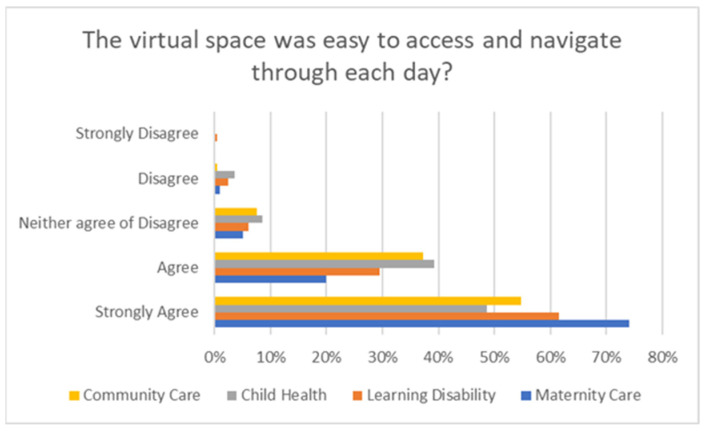
Ease of navigating the virtual space graph.

**Figure 5 nursrep-15-00061-f005:**
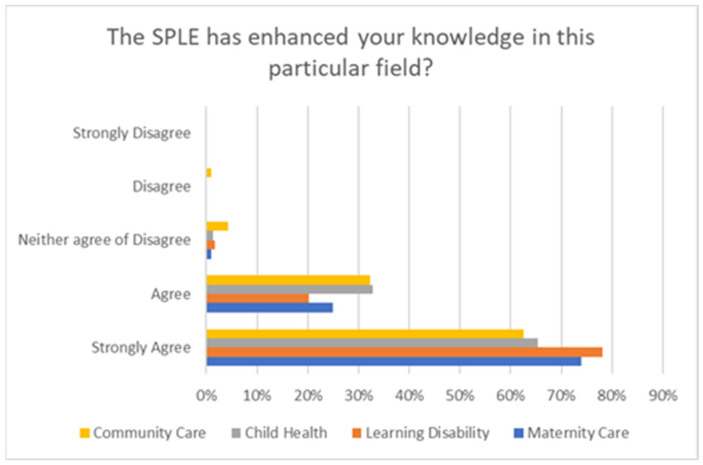
Impact of SPLE on knowledge enhancement in each field graph.

**Table 1 nursrep-15-00061-t001:** Numbers of students who completed SPLE evaluation questionnaires per week.

SPLE Week	Completed Questionnaire
Maternity Care	216
Leaning Disability	205
Child Health	198
Community Care	199

## Data Availability

Students provided consent for their data to be used for evaluative purposes with a view to inclusion of future publication only.

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
