# Peer review of "Simulated Practice Learning Experience in a Virtual Environment: An Innovative Pedagogical Approach to Practice Learning for Nursing Students"

_nursrep, 2025, doi:10.3390/nursrep15020061_

Round 1

Reviewer 1 Report

Comments and Suggestions for Authors

Dear authors,

Firstly, I would like to congratulate you on the initiative to carry out a study with this type of interactive program.

One of the aspects that caught my attention the most has to do with the title. On the one hand because I don't think you use simulation (with its 3 phases) and on the other hand I question the use of the concept of virtual environment, since virtual learning can include immersive environments with virtual reality devices (which I don't think have been used). Will it really be virtual or online or B-learning?

I have attached a file with my comments and suggestions for revising the article.

Reviewer 2 Report

Comments and Suggestions for Authors

The article titled "Simulated Practice Learning Experience in a Virtual Environment: An Innovative Pedagogical Approach to Practice Learning for Nursing Students" was considered for review. The authors implemented an innovative and alternative approach to practice learning for a cohort of first-year adult and mental health nursing students. The research focus analyzed in the manuscript is relevant to the improvement of the teaching-learning process in nursing. However, it lacks alignment with the application of simulation techniques. The manuscript demonstrates limitations in producing evidence and in the theoretical-methodological and analytical consistency of the discussion of the data. Below are some additional comments:

Introduction: The explanation of the research focus and formulation of the guiding question is insufficiently supported. It mainly emphasizes the importance of the Simulated Practice Learning Experience due to changes brought about by COVID-19 but lacks proper argumentation regarding the effectiveness of clinical simulation as structured in the study. Furthermore, the research objective is not clearly presented. A scoping review of the literature from the past five years is mentioned as having been conducted to gather relevant evidence on virtual practice learning, yet no references are provided.

Methodology: The methodological framework is unclear and does not sufficiently outline the theoretical context for structuring the intervention proposal (Simulated Practice Learning Experience). It fails to ensure reproducibility, as it does not align with criteria of impartiality required for scientific research. While the manuscript highlights several positive reflections, these are only briefly addressed in the results.

Results: The reference to the impact of the proposal is inconsistent, as the data presented do not clearly demonstrate, either in the realm of theoretical knowledge or practical skills, significant transformations triggered by the Simulated Practice Learning Experience.

Conclusions: The conclusions lack depth, are vague, and fail to address the research objectives in a concrete manner. They do not provide substantial contributions to nursing education, practice, or research.

Reviewer 3 Report

Comments and Suggestions for Authors

The article addresses the effects of simulated learning environments on nursing students by introducing an innovative pedagogy approach. The topic is quite timely and important in the context of the challenges brought about by the COVID-19 pandemic. However, some sections of the text lack depth and structure.

Abstract

The purpose and methods of the study are clearly stated.

The implications summarize the effects on students.

The limitations of the study are not stated in the abstract; their inclusion would provide better context for readers.

The abstract could be revised to be more concise and targeted.

Introduction

The context of the study is well related to the impact of COVID-19.

The gaps in nursing education and the contribution of this study are well explained based on the literature.

The sources are up-to-date and sufficient, and could be summarized and shortened a bit more.

The research questions are not present.

Methodology

The use of mixed methods makes the results stronger and more reliable.

The evaluation of student feedback provided a good source of data.

The quantitative and qualitative data collection tools need to be explained in more detail. In particular, information on the validity and reliability of the questionnaires used should be provided.

It can be explained more clearly how the sample size was determined and which groups it covers.

Findings

Data are supported with visuals and analyses are presented in an understandable manner.

Both quantitative and qualitative results are presented in a balanced manner.

Discussion

Findings are well discussed and related to the literature.

Contributions of SPLE (Simulated Practice Learning Experience) to nursing education are clarified.

Discussion section includes repetitions of findings in some places. A more concise discussion can be made.

Contradictions or gaps stated in the literature can be addressed more clearly.

Conclusions and recommendations

Both strengths and contributions of the study are summarized.

Recommendations for future research are presented.

The limitations of the study can be addressed more comprehensively in the conclusion section. For example, the effect of the fact that it was conducted with only first-year students on general applicability can be discussed.

More comprehensive recommendations can be provided, especially for educational institutions.

General

Writing style is generally understandable and fluent.

Article format and headings are organized properly.

Round 2

Reviewer 1 Report

Comments and Suggestions for Authors

Dear authors,

Thank you for your care in integrating the suggestions made and the changes made to the article that have made it clearer and more sustained.

I have attached (see below) a file with comments to finalize the revision of the article.

Comments on the Quality of English Language

I would suggest revising the formatting of the article (the version I read was unformatted) and the English language, in particular the agreement of verbs with the subject.

Reviewer 2 Report

Comments and Suggestions for Authors

The article titled "Simulated Practice Learning Experience in a Virtual Environment: An Innovative Pedagogical Approach to Practice Learning for Nursing Students" was considered for review. The authors implemented an innovative and alternative approach to practice learning for a cohort of first-year adult and mental health nursing students. I think that the authors have adequately addressed the comments made by the reviewers in the revised version of the manuscript. Therefore, I have no further comments.

Author Response

Thanks for your review and sign off.